

# Comparative efficacy of gas therapy for diabetic foot ulcers using network meta-analysis

Jing Yang[1,*], Peng Ning[1,*], Jiali Huang[1], Hong Ouyang[1], Jiaxing Zhang[1], Fan Yang[1], Hongyi Cao[1] and Fan Zhang[2]

[1] Department of Endocrine and Metabolism, Chengdu Fifth People's Hospital (The Second Clinical Medical College, Affiliated Fifth People's Hospital of Chengdu University of Traditional Chinese Medicine), Geriatric Diseases Institute of Chengdu/Cancer Prevention and Treatment Institute of Chengdu, Chengdu, Sichuan, China

[2] Department of Endocrine and Metabolism, Guang'an People's Hospital (Sichuan University West China Hospital Guang'an Hospital), Guang'an, Sichuan, China

[*] These authors contributed equally to this work.

Corresponding authors
Peng Ning, ningpeng_yun@163.com
Fan Zhang, 307562489@qq.com

## ABSTRACT

**Objective.** Diabetic foot ulcers (DFUs) pose significant clinical challenges, with gas therapy emerging as a promising intervention. However, the comparative efficacy of various gas therapy modalities remains unclear. This study evaluates the effectiveness of different gas therapies, particularly hyperbaric oxygen therapy (HBOT), in improving DFU outcomes.

**Methods.** We searched three major databases, PubMed, Embase, and the Cochrane Library, for randomized controlled trials (RCTs) published up to March 3, 2024, assessing the efficacy of different gas therapies in managing DFUs. Primary outcomes included ulcer healing and area reduction rates, while secondary outcomes encompassed healing time, amputation rate, and adverse events. A network meta-analysis was performed using R, with surface under the cumulative ranking curve (SUCRA) values calculated to rank therapies.

**Results.** A total of 34 RCTs involving 2,268 DFUs were included in this analysis. HBOT ranked highest for the healing rate (SUCRA = 0.8 14) and area reduction rate (SUCRA = 0.730) but also had a higher amputation rate (SUCRA = 0.621). Except for carbon dioxide therapy, HBOT demonstrated significantly greater healing rates than standard of care (SOC) (mean difference (MD) = −2.71, 95% confidence interval (CI) [−4.85 to −1.34]) and topical oxygen therapy (MD = −2.03, 95% CI [−4.50 to −0.32]), while pairwise comparisons among other gaseous therapies were non-significant ($P > 0.05$). For wound area reduction, HBOT was superior to SOC (MD = 0.39, 95% CI [0.11–0.67]), whereas differences among other gaseous therapies remained non-significant ($P > 0.05$). There was substantial heterogeneity in the area reduction rate in net work analysis ($I^2 = 87\%$). Subgroup analyses revealed greater area reduction in DFUs treated with HBOT when the treatment duration exceeded six weeks.

**Conclusions.** HBOT improves ulcer healing and area reduction rates in DFU patients; however, its association with higher amputation rates warrants cautious use, considering resource availability. Given the limited number and quality of included studies, further high-quality research is needed to validate these findings.

## INTRODUCTION

With the rising prevalence of diabetes, the incidence of diabetic foot ulcers (DFUs) is also increasing. The International Diabetes Foundation estimates that DFUs affect 40–60 million people worldwide (*McDermott et al., 2023*). If left untreated, DFUs can progress to soft tissue infections, gangrene, and even amputation (*Primadhi et al., 2023*). Moreover, DFUs are associated with mortality rates of up to 50% within five years (*Chen et al., 2023b*), posing a significant health threat and placing a substantial burden on society. DFUs typically result from prolonged hyperglycemia, neuropathy, and vascular disease (*Armstrong et al., 2023*). Traditional treatments, including wound debridement, antibiotic regimens, and advanced wound dressings, remain fundamental but have notable limitations. For instance, antibiotics are increasingly hindered by antimicrobial resistance and poor tissue penetration in ischemic environments, while debridement often requires repeated procedures and does not address underlying vascular insufficiency. Similarly, wound dressings may struggle to maintain optimal moisture balance or oxygenation in severe ulcers. These limitations underscore the need for adjuvant therapies to enhance healing outcomes.

The advancement of gas therapy techniques and other clinical application of therapies such as hyperbaric oxygen therapy (HBOT), topical oxygen therapy (TOT), ozone, carbon dioxide ($CO_2$), nitric oxide (NO), and cold atmospheric plasma (CAP) therapies offer promising solutions to these challenges. For example, HBOT enhances tissue oxygenation in hypoxic wounds (*Wenhui et al., 2021*; *Sharma et al., 2021*), TOT delivers localized oxygen to stimulate granulation (*Yellin et al., 2022*), ozone exhibits potent antimicrobial and anti-inflammatory properties (*Astasio-Picado et al., 2023*), $CO_2$ improves microcirculation (*Finžgar et al., 2021*), NO modulates vasodilation and inflammation (*Walton, Minton & Cook, 2019*), and CAP promotes biofilm disruption and cell proliferation (*Barjasteh et al., 2023*). By targeting hypoxia, infection, and impaired angiogenesis, these therapies address key limitations of conventional treatments. Studies indicate that gas therapy may enhance wound healing by improving oxygen supply, controlling infection, and promoting angiogenesis and tissue repair (*Kwee et al., 2024*; *Frykberg et al., 2023*; *Kushmakov et al., 2018*; *Chen et al., 2023a*; *Ma et al., 2023*). In May 2023, the International Working Group on the Diabetic Foot (IWGDF) updated its guidelines to conditionally recommend HBOT and TOT as adjuncts for refractory neuro-ischemic or ischemic DFUs (*Chen et al., 2024*). However, the relative efficacy of various gas-therapy modalities remains insufficiently studied.

To address this gap, we conducted a network meta-analysis (NMA) to compare the relative efficacy of gas therapies for DFUs. Unlike conventional pairwise meta-analyses, NMA integrates direct and indirect evidence to enable simultaneous comparison of multiple interventions, enhancing statistical power and ranking therapeutic options. We hypothesize that oxygen-based therapies (HBOT, TOT) will demonstrate superior

ulcer healing compared to non-oxygen modalities (*e.g.*, ozone, CAP) due to their direct oxygenation benefits, though differences in accessibility and side-effect profiles may influence clinical applicability. The findings of this NMA may provide clinicians with additional evidence to guide treatment selection for DFU management and support the further development and application of gas therapy in clinical practice.

## METHODS

### Registration

This NMA was conducted in accordance with the Preferred Reporting Items for Systematic Evaluation and Meta-Analysis statement (*Page et al., 2021*). The study protocol is registered with the International Prospective Systematic Evaluation Register (CRD42024504767).

### Search strategy

Electronic databases, including PubMed, Embase, and Cochrane, were systematically searched. The search strategy was structured using the PICOS tool: population: patients with DFUs; intervention: HBOT and TOT as well as ozone, NO, $CO_2$, and CAP therapy; and study type: RCT. A complete list of search terms is provided in Table S1. In addition, reference lists of previous systematic reviews and meta-analyses were screened to identify any missing studies. The retrieved literature was managed using EndNote version X9 (Thomson ResearchSoft, Stanford, CA, USA).

### Research selection

The inclusion criteria were as follows: (1) patients with a documented diagnosis of diabetes mellitus (type 1, type 2, or other forms, excluding gestational diabetes) based on the American Diabetes Association standards for the diagnostic criteria (*ElSayed et al., 2023*), with DFUs diagnosed according to the IWGDF 2023 guidelines (*Van Netten et al., 2024*), regardless of nationality or race; (2) interventions involving any gas therapy modality, including HBOT, TOT, or ozone, $CO_2$, NO, or CAP therapy; (3) control measures including standard of care (SOC) or hyperbaric air therapy (HBAT), where SOC includes blood sugar control, infection control, sharp device debridement, and routine dressing change for patients, while HBAT refers to the supply of air instead of oxygen under high-pressure conditions as part of SOC; and (4) study type restricted to RCTs. The exclusion criteria were: (1) reviews, systematic evaluations, conference abstracts, retrospective studies, cross-sectional studies, and cross-over RCTs; (2) studies lacking relevant outcome measures or with data that could not be extracted; and (3) articles not published in English.

The selection process was conducted in two phases. In the first phase, two researchers (J Huang and H Ouyang) independently screened study titles and abstracts to identify potentially eligible articles. In the second phase, full-text articles of the selected studies were thoroughly reviewed by the same researchers to confirm eligibility. Any disagreements between the reviewers were resolved through discussion or adjudication by a third researcher (P Ning).

## Data extraction

Data extraction was carried out based on predetermined outcome measures. Two authors (J Huang and H Ouyang) independently extracted data from the included studies using preset tables in Excel (version 2021, Microsoft, Redmond, WA, USA). Extracted data included study author, year of publication, country, sample size, duration of treatment, treatment details (various types of gas therapy and SOC), primary outcomes (ulcer healing rate and ulcer area reduction rate), and secondary outcome (ulcer healing time, amputation rate (including major and minor amputations), and adverse events) were included. If a study reported multiple outcome measures for different treatment durations, data from the longest treatment duration were selected. For RCTs presenting both per protocol and intention to treat (ITT) measures, ITT data were used. To maintain consistency and comparability, RCTs with missing data often reported outcome ratios based on the initial sample size. When outcome metrics were available only as images, numerical data were extracted using GetData Graph Digitizer (version 2.25; GetData Software Development Company, Sydney, Australia). Discrepancies in data extraction between the two researchers were resolved through adjudication by a third researcher (F Yang).

## Risk-of-bias assessment

The risk of bias was assessed using the Cochrane Risk of Bias Tool (version 2.0), including random sequence generation (selection bias), allocation concealment (selection bias), blinding of participants and personnel (performance bias), blinding of outcome assessment (detection bias), incomplete outcome data (attrition bias), selective reporting (reporting bias), and other potential biases. Study quality was evaluated as low risk, unclear risk, and high risk. Two researchers (J Zhang and J Yang) independently conducted the assessments, with discrepancies resolved by a third researcher (F Yang).

## Data analysis

Statistical analyses were performed using R software (version 4.3.1; R Core Team, Vienna, Austria), with Bayesian network meta-analysis implemented *via* the "gemtc" and "rjags" packages (*Shim et al., 2019*). GraphPad Prism (version 9.4.1; GraphPad Software, San Diego, CA, USA) was used for image mapping and plotting. Bayesian methods were selected for their ability to incorporate prior distributions, directly estimate treatment ranking probabilities (*via* surface under the cumulative ranking curve (SUCRA) values), and flexibly handle sparse or complex network structures compared to frequentist approaches. First, a network graph was constructed using R to show all available evidence for each intervention as a simple overview. Second, Bayesian network analysis based on Markov chain Monte Carlo (*Shim et al., 2019*) was used to analyze the efficacy of different gas treatments on the primary and secondary outcome metrics in patients with DFUs. The number of tuning and simulation iterations was 5,000 and 20,000, respectively. Study heterogeneity was assessed using the $I^2$ value. A fixed-effects model was applied when $I^2 \leq 50\%$, indicating low or negligible heterogeneity, while a random-effects model was used for $I^2 > 50\%$, reflecting substantial heterogeneity. The results of the generated ranking tables are expressed as mean difference (MD) and 95% confidence interval (CI). The data were not

statistically significant when the 95% CI value contained 0. Next, to assess the probability that each intervention was most effective, the SUCRA value was calculated (values ranging from 0–1). Higher SUCRA values indicate a greater likelihood that a treatment is effective or has the highest effectiveness, thereby maximizing the potential to achieve the best outcome metrics (*Mbuagbaw et al., 2017*). Then, the primary outcome metrics of all included studies were analyzed for heterogeneity; the smaller the $I^2$ value, the lesser the heterogeneity. Thus, 0%–25% was considered no heterogeneity, 25%–50% was considered mild heterogeneity, 50%–75% was considered moderate heterogeneity, and 75%–100% was considered high heterogeneity (*Higgins et al., 2023*). Subgroup analyses were used to explore the source of heterogeneity for outcomes with high heterogeneity. Finally, publication bias for the primary outcomes was assessed using comparison-adjusted funnel plots in Stata, version 15.0 (StataCorp LLC, College Station, TX, USA). A significance level of $\alpha = 0.05$ was used to determine statistical significance.

# RESULTS

## Study selection and characteristics

A total of 255 articles were identified in our initial database search, and 157 were selected after accounting for duplicates. Four articles were excluded because they were not in English, 92 were excluded because they were not RCTs or cross-over RCTs, and 33 were excluded because they did not meet the PICOS principles. In addition to the database search, the reference lists of previous systematic reviews and meta-analyses in the field were searched, and six articles that met the inclusion criteria were identified. Finally, 34 RCTs (Fig. 1) containing 2,268 cases of DFUs were included. The details of the included RCTs are presented in Table 1. HBOT, TOT, topical hyperbaric oxygen therapy (THOT), ozone therapy, oxygen-ozone therapy (OOT), $CO_2$ therapy, NO therapy, and CAP therapy were the included gas therapies.

## Risk of bias in studies

The risk of bias was assessed for all included RCTs. Among the 34 included RCTs, 12 did not explicitly describe the generation of randomized sequences, and 14 RCTs lacked details on allocation concealment. Moreover, 24 RCTs did not specify the blinding protocol for participants and personel, while 25 RCTs did not clearly outline blinding procedures for outcome assessment. Four RCTs had incomplete data, 18 exhibited selective reporting, and 10 had potential other biases. Figure 2 shows an overview of the risk of bias at both individual and overall levels.

## Main outcome indicators
### Healing rate

A total of 21 RCTs examined the effect of different gas treatments on the healing rate of DFUs, with the reticulation diagram shown in Fig. 3A. The heterogeneity test suggested that $I^2 = 62\%$; thus, a random-effects model was used. Although SUCRA analysis (cumulative probability ranking graph, Fig. 4A) ranked $CO_2$ therapy as the most effective therapy for DFUs healing (SUCRA = 0.998), this result is derived from a single RCT (*Macura et al.,*

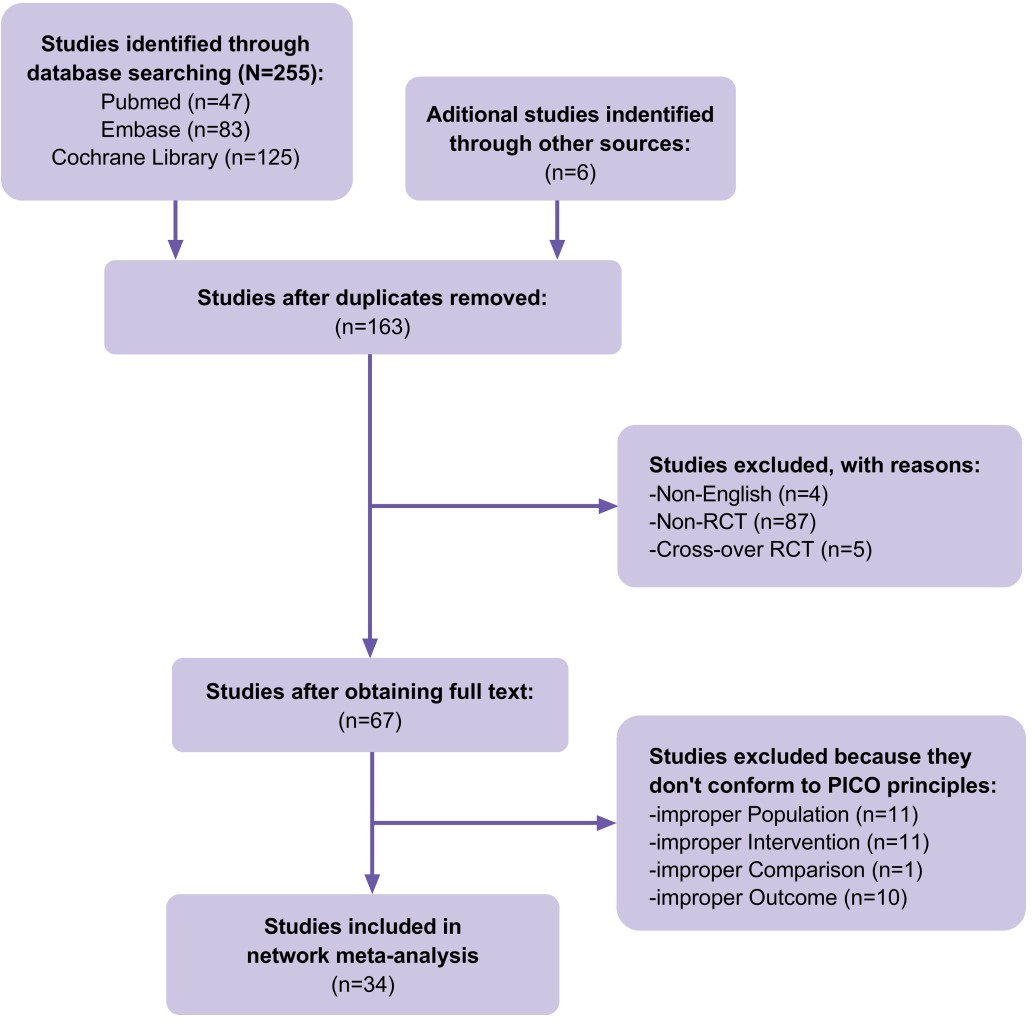

Figure 1  Flow of trials through the review.

*2020*), suggesting potential bias due to the limited sample size. Conversely, HBOT, ranked second (SUCRA = 0.814), is supported by multiple RCTs, providing more robust evidence. The league table (Table 2A) showed that HBOT had a higher healing rate than SOC (MD = −2.71, 95% CI [−4.85 to −1.34]) and TOT (MD = −2.03, 95% CI [−4.50 to −0.32]). Pairwise comparison among other gas therapy modalities showed no statistically significant differences, with 95% CIs containing 0 ($P > 0.05$). Forest plots illustrating healing rates for each gas therapy compared to SOC are shown in Fig. 5A.

### Reduction rate of DFU area

A total of 22 RCTs examined the effect of different gas treatments on the reduction rate of DFU area, and the network diagram is shown in Fig. 3B. The heterogeneity test suggested that $I^2 = 87\%$; thus, a random-effects model was used. Similar to the healing rate findings, SUCRA analysis (cumulative probability ranking graph Fig. 4B) ranked $CO_2$ therapy highest (SUCRA = 0.828). However, this result must also be interpreted with caution

**Table 1  Characteristics of RCTs about the efficacy of gas therapy in patients with diabetic foot ulcers.**

| Study | Region | Wagner grading | Intervention | | | Comparison | | | Treatment time | Outcome |
|---|---|---|---|---|---|---|---|---|---|---|
| | | | N | Ulcer area (cm²) | Method | N | Ulcer area (cm²) | Method | | |
| Doctor, 1992 (*Doctor, Pandya & Supe, 1992*) | India | NA | 15 | NA | HBOT | 15 | NA | SOC | 2 weeks | 4 |
| Faglia, 1996 (*Faglia et al., 1996*) | Italy | 2–4 | 35 | NA | HBOT | 33 | NA | SOC | NA | 4,5 |
| Abidia, 2002 (*Abidia et al., 2003*) | UK | 1–2 | 8 | 2.5 ± 2.8 | HBOT | 8 | 2.4 ± 3.0 | HBAT | 6 weeks | 1,2,4,6,7 |
| Kessler, 2003 (*Kessler et al., 2003*) | France | 1–3 | 14 | 2.3 ± 2.2 | HBOT | 13 | 2.8 ± 2.4 | SOC | 2 weeks | 1,2, |
| Duzgun, 2008 (*Duzgun et al., 2008*) | Turkey | 2–4 | 50 | NA | HBOT | 50 | NA | SOC | 20–30 days | 1,4 |
| Löndahl, 2010 (*Löndahl et al., 2010*) | Sweden | 2–4 | 38 | 7.5 ± 12.2 | HBOT | 37 | 6.0 ± 8.7 | HBAT | 8–10 weeks | 1,4,5 |
| Ma, 2013 (*Ma et al., 2013*) | China | 1–3 | 18 | 4.2 ± 1.0 | HBOT | 18 | 4.4 ± 1.0 | SOC | 2 weeks | 1,2, |
| Fedorko, 2016 (*Fedorko et al., 2016*) | USA | 2–4 | 49 | 3.8 ± 4.8 | HBOT | 54 | 3.6 ± 5.7 | HBAT | 12 weeks | 1,2,4,5 |
| Chen, 2017 (*Chen et al., 2017*) | China | 1–3 | 20 | NA | HBOT | 18 | NA | SOC | 4 weeks | 1,4,6 |
| Santema, 2018 (*Santema et al., 2018*) | Netherlands | 2–4 | 60 | NA | HBOT | 60 | NA | SOC | 8 weeks | 1,4,5 |
| Perren, 2018 (*Perren et al., 2018*) | Malta | 1–3 | 13 | 11.7 ± 2.2 | HBOT | 13 | 10.6 ± 0.9 | SOC | 4 weeks | 2 |
| Dhamodharan, 2019 (*Dhamodharan et al., 2019*) | India | 3–4 | 15 | NA | HBOT | 17 | NA | SOC | 20 days | 2 |
| Salama, 2019 (*Salama et al., 2019*) | Egypt | 2–3 | 15 | 7.8 ± 4.0 | HBOT | 15 | 8.4 ± 4.2 | SOC | 2 months | 1,2,4,5 |
| Kumar, 2020 (*Kumar et al., 2020*) | India | 2–4 | 28 | 2.9 ± 1.5 | HBOT | 26 | 3.0 ± 2.8 | SOC | 6 weeks | 1,3,4 |
| Wadee, 2021 (*Wadee, Fahmy & El-Deen, 2021*) | Egypt | 2 | 25 | 5.5 ± 0.7 | HBOT | 25 | 5.6 ± 0.8 | SOC | 6 weeks | 2 |
| Yu, 2016 (*Yu et al., 2016*) | Canada | NA | 10 | 1.4 ± 1.0 | TOT | 10 | 1.7 ± 1.3 | SOC | 8 weeks | 1,2 |
| Driver, 2017 (*Driver et al., 2017*) | USA | NA | 61 | 2.0 ± 1.7 | TOT | 61 | 2.3 ± 1.7 | SOC | 12 weeks | 1,4,5 |
| Niederauer, 2018 (*Niederauer et al., 2018*) | USA | NA | 74 | 3.5 ± 1.7 | TOT | 72 | 3.9 ± 2.0 | SOC | 12 weeks | 1,3,5 |
| Frykberg, 2020 (*Frykberg et al., 2020*) | Multinational | NA | 36 | 3.0 ± 2.7 | TOT | 37 | 3.2 ± 2.5 | SOC | 12 weeks | 1,2,3,4,5,6 |
| Serena, 2021 (*Serena et al., 2021*) | USA | 1–3 | 81 | 2.9 ± 2.9 | TOT | 64 | 3.5 ± 4.1 | SOC | 12 weeks | 1,2,5 |
| Wang, 2021 (*Wang et al., 2021*) | China | NA | 29 | NA | TOT | 29 | NA | SOC | NA | 3 |
| He, 2021 (*He et al., 2021*) | China | 2–3 | 40 | 39.6 ± 23.4 | TOT | 40 | 42.1 ± 38.6 | SOC | 8 weeks | 1,2,3,4 |
| Anirudh, 2021 (*Anirudh et al., 2021*) | India | 2–3 | 10 | 20.8 ± 17.0 | TOT | 9 | 17.3 ± 15.0 | SOC | 6 weeks | 2,4,5,6 |
| Leslie, 1988 (*Leslie et al., 1998*) | USA | NA | 12 | 5.5 ± 5.5 | THOT | 16 | 3.2 ± 2.6 | SOC | 2 weeks | 2 |
| Martinez-Sanchez, 2005 (*Martínez-Sánchez et al., 2005*) | Cuba | NA | 51 | 58.0 ± 0.5 | OT | 49 | 54.8 ± 0.4 | SOC | 20 days | 1,2,3 |
| Izadi, 2018 (*Izadi et al., 2019*) | Iran | 1–4 | 100 | 13.4 ± 14.1 | OT | 100 | 12.7 ± 0.9 | SOC | NA | 4 |
| Wainstein, 2011 (*Wainstein et al., 2011*) | Lsrael | 2–4 | 32 | 4.9 ± 4.4 | OOT | 29 | 3.5 ± 3.8 | SOC | 12 weeks | 1,2,5 |
| Zhang, 2014 (*Zhang et al., 2014*) | China | 2–4 | 25 | 11.7 ± 0.7 | OOT | 25 | 10.8 ± 0.9 | SOC | 20 days | 1,2, |
| Macura, 2020 (*Macura et al., 2020*) | Slovenia | NA | 30 | 4.9 ± 7.1 | CO2 | 27 | 4.1 ± 4.5 | SOC | 4 weeks | 1,2 |
| Edmonds, 2018 (*Edmonds et al., 2018*) | UK | NA | 75 | 2.1 ± 3.3 | NOT | 73 | 1.6 ± 2.2 | SOC | 12 weeks | 1,2,4,5 |
| Mirpour, 2020 (*Mirpour et al., 2020*) | Iran | 2 | 22 | 3.5 ± 4.2 | CAP | 22 | 2.1 ± 1.5 | SOC | 3 weeks | 2 |
| Stratmann, 2020 (*Stratmann et al., 2020*) | Germany | 1–2 | 33 | NA | CAP | 32 | NA | SOC | 2 weeks | 5,6 |
| Samsavar, 2021 (*Samsavar et al., 2021*) | Iran | NA | 10 | 7.9 ± 10.2 | CAP | 10 | 5.6 ± 7.8 | SOC | 6 weeks | 2,5 |
| Hiller, 2022 (*Hiller et al., 2022*) | Germany | 1–2 | 14 | NA | CAP | 13 | NA | SOC | 2 weeks | 2 |

**Notes.**

HBOT, Hyperbaric oxygen therapy;  HBAT, Hyperbaric air therapy;  TOT, Topical oxygen therapy;  THOT, Topical hyperbaric oxygen therapy;  OT, Ozone therapy;  OOT, Oxygen-Ozone therapy;  $CO_2$, Carbon dioxide;  NOT, Nitric oxide therapy;  CAP, Cold atmospheric plasma;  SOC, Standard of care.

1. Healing rate; 2. Area reduction rate; 3. Healing time; 4. Amputation rate; 5. Adverse event; 6. Quality of life; 7. Cost of treatment.

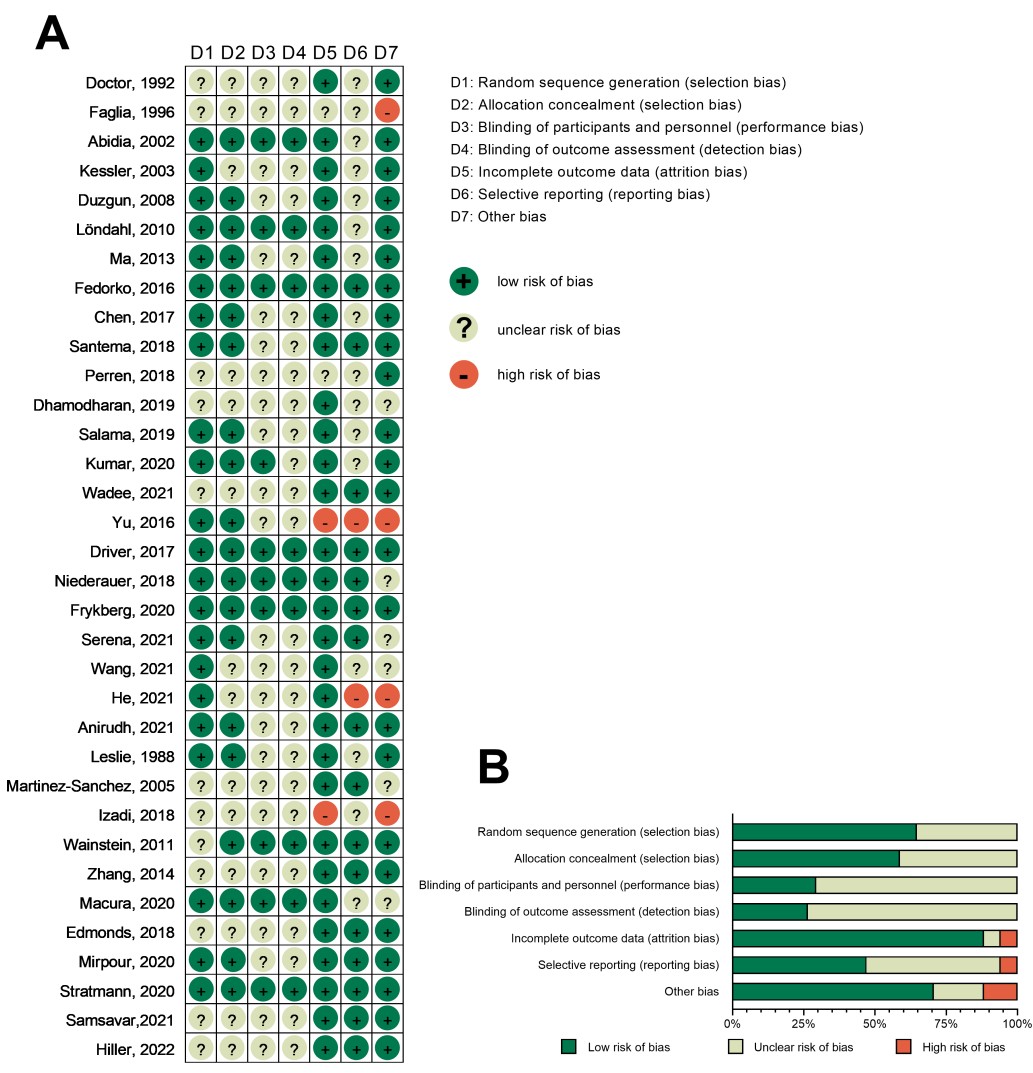

**Figure 2 Risk-of-bias graph.** (A) Risk-of-bias summary: review authors' judgments about each risk-of-bias item for each included study. (B) Risk-of-bias graph: judgments about each risk-of-bias item presented as percentages across all included studies. Note: *Doctor, Pandya & Supe, 1992*; *Faglia et al., 1996*; *Abidia et al., 2003*; *Kessler et al., 2003*; *Duzgun et al., 2008*; *Löndahl et al., 2010*; *Ma et al., 2023*; *Fedorko et al., 2016*; *Chen et al., 2017*; *Santema et al., 2018*; *Perren et al., 2018*; *Dhamodharan et al., 2019*; *Salama et al., 2019*; *Kumar et al., 2020*; *Wadee, Fahmy & El-Deen, 2021*; *Yu et al., 2016*; *Driver et al., 2017*; *Niederauer et al., 2018*; *Frykberg et al., 2020*; *Serena et al., 2021*; *Wang et al., 2021*; *He et al., 2021*; *Anirudh et al., 2021*; *Leslie et al., 1988*; *Martínez-Sánchez et al., 2005*; *Izadi et al., 2018*; *Wainstein et al., 2011*; *Zhang et al., 2014*; *Macura et al., 2020*; *Edmonds et al., 2018*; *Mirpour et al., 2020*; *Stratmann et al., 2020*; *Samsavar et al., 2021*; *Hiller et al., 2022*.

because of the inclusion of only one RCT (*Macura et al., 2020*). Meanwhile, HBOT ranked second (SUCRA = 0.730) and was supported by a broader evidence base. The league table (Table 2B) indicated that HBOT achieved a higher ulcer area reduction rate than SOC (MD = 0.39, 95% CI [0.11–0.67]). However, pairwise comparisons among other gas therapies showed no statistically significant differences in ulcer area reduction rates ($P > 0.05$). The

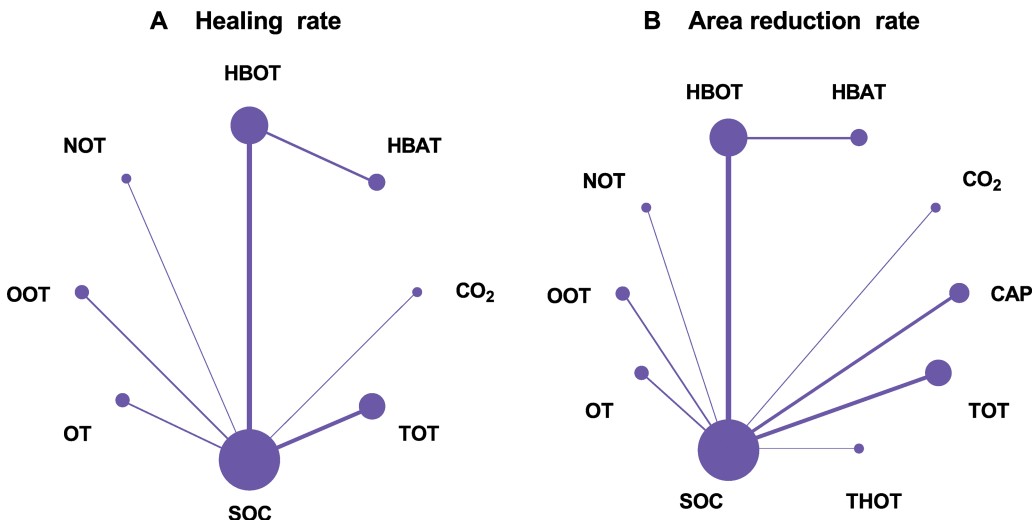

**Figure 3** **Network plots.** (A) Healing rate; (B) Area reduction rate. HBOT, Hyperbaric oxygen therapy; HBAT, Hyperbaric air therapy; TOT, Topical oxygen therapy; THOT, Topical hyperbaric oxygen therapy; OT, Ozone therapy; OOT, Oxygen-Ozone therapy; $CO_2$, Carbon dioxide; NOT, Nitric oxide therapy; CAP, Cold atmospheric plasma; SOC, Standard of care.

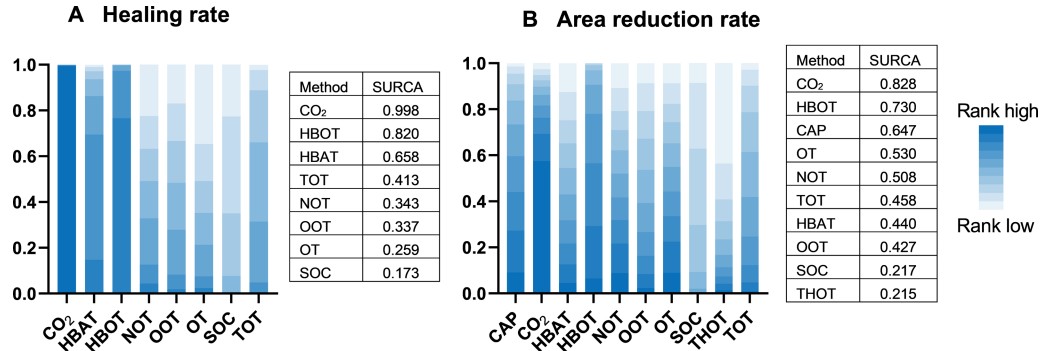

**Figure 4** **Cumulative probability ranking graph.** (A) Healing rate; (B) Area reduction rate. HBOT, Hyperbaric oxygen therapy; HBAT, Hyperbaric air therapy; TOT, Topical oxygen therapy; THOT, Topical hyperbaric oxygen therapy; OT, Ozone therapy; OOT, Oxygen-Ozone therapy; $CO_2$, Carbon dioxide; NOT, Nitric oxide therapy; CAP, Cold atmospheric plasma; SOC, Standard of care.

forest plot illustrating ulcer-area reduction rates for each gas treatment compared to SOC is shown in Fig. 5B.

## Secondary outcome indicators
### *Healing time*

A total of six RCTs evaluated the effect of different gas treatments on DFUs healing time, as depicted in the network diagram (Fig. S1A). The heterogeneity test suggested that $I^2 =$ 73%; therefore, a random-effects model was used. SUCRA analysis (cumulative probability ranking graph Fig. S2A) ranked HBOT treatment highest for DFU healing time (SUCRA =

**Table 2 Mean difference (MD) and 95% confidence interval (CI) for primary outcome measure.** (A) Healing rate; (B) Area reduction rate.

**(A) Healing rate**

| $CO_2$ | | | | | | | |
|---|---|---|---|---|---|---|---|
| 46.91 (4.76, 191.30) | **NOT** | | | | | | |
| 46.94 (4.85, 191.37) | 0.01 (−3.55, 3.53) | **OOT** | | | | | |
| 47.27 (5.10, 191.76) | 0.35 (−3.63, 4.35) | 0.32 (−3.13, 3.84) | **OT** | | | | |
| 47.37 (5.34, 191.80) | 0.45 (−2.40, 3.30) | 0.43 (−1.64, 2.54) | 0.10 (−2.69, 2.87) | **SOC** | | | |
| 46.70 (4.65, 191.00) | −0.24 (−3.34, 2.85) | −0.25 (−2.65, 2.15) | −0.58 (−3.61, 2.43) | −0.68 (−1.87, 0.47) | **TOT** | | |
| 45.21 (3.14, 189.97) | −1.53 (−5.62, 1.81) | −1.56 (−5.10, 1.33) | −1.87 (−5.93, 1.43) | −1.99 (−4.74, 0.08) | −1.30 (−4.31, 1.06) | **HBAT** | |
| 44.48 (2.47, 189.08) | −2.25 (−6.01, 0.65) | −2.28 (−5.38, 0.08) | −2.59 (−6.30, 0.26) | **−2.71 (−4.85, −1.34)** | **−2.03 (−4.50, −0.32)** | −0.72 (−2.50, 0.94) | **HBOT** |

**(B) Area reduction rate**

| **CAP** | | | | | | | | | |
|---|---|---|---|---|---|---|---|---|---|
| −0.35 (−1.35, 0.69) | **$CO_2$** | | | | | | | | |
| 0.18 (−0.59, 0.97) | 0.53 (−0.58, 1.63) | **HBAT** | | | | | | | |
| −0.05 (−0.58, 0.51) | 0.29 (−0.66, 1.24) | −0.23 (−0.79, 0.33) | **HBOT** | | | | | | |
| 0.12 (−0.71, 0.98) | 0.46 (−0.69, 1.61) | −0.06 (−1.01, 0.88) | 0.17 (−0.59, 0.93) | **NOT** | | | | | |
| 0.19 (−0.47, 0.90) | 0.54 (−0.49, 1.59) | 0.01 (−0.78, 0.82) | 0.24 (−0.31, 0.82) | 0.08 (−0.79, 0.95) | **OOT** | | | | |
| 0.10 (−0.72, 0.93) | 0.44 (−0.70, 1.57) | −0.08 (−1.01, 0.84) | 0.15 (−0.59, 0.88) | −0.02 (−1.00, 0.96) | −0.10 (−0.94, 0.74) | **OT** | | | |
| 0.34 (−0.12, 0.82) | 0.68 (−0.22, 1.59) | 0.16 (−0.47, 0.79) | **0.39 (0.11, 0.67)** | 0.22 (−0.49, 0.93) | 0.14 (−0.36, 0.63) | 0.24 (−0.44, 0.92) | **SOC** | | |
| 0.44 (−0.38, 1.28) | 0.78 (−0.37, 1.91) | 0.25 (−0.67, 1.18) | 0.49 (−0.26, 1.23) | 0.32 (−0.67, 1.31) | 0.24 (−0.61, 1.09) | 0.34 (−0.64, 1.32) | 0.10 (−0.59, 0.79) | **THOT** | |
| 0.17 (−0.40, 0.74) | 0.51 (−0.47, 1.47) | −0.01 (−0.73, 0.69) | 0.22 (−0.23, 0.64) | 0.05 (−0.75, 0.82) | −0.03 (−0.65, 0.56) | 0.07 (−0.70, 0.82) | −0.17 (−0.52, 0.16) | −0.27 (−1.05, 0.49) | **TOT** |

**Notes.**

HBOT, Hyperbaric oxygen therapy; HBAT, Hyperbaric air therapy; TOT, Topical oxygen therapy; THOT, Topical hyperbaric oxygen therapy; OT, Ozone therapy; OOT, Oxygen-Ozone therapy; $CO_2$, Carbon dioxide; NOT, Nitric oxide therapy; CAP, Cold atmospheric plasma; SOC, Standard of care.
Statistically significant differences are highlighted in bold.

0.915). However, the league table (Table S2A) showed no statistically significant differences in healing time among gas therapy modalities ($P > 0.05$). Forest plots comparing each gas treatment with SOC for ulcer healing time are shown in Fig. S3A.

*Amputation rate*

A total of 16 RCTs investigated the effect of different gas treatments on the amputation rate of DFUs, as illustrated in the network diagram shown in Fig. S1B. The heterogeneity

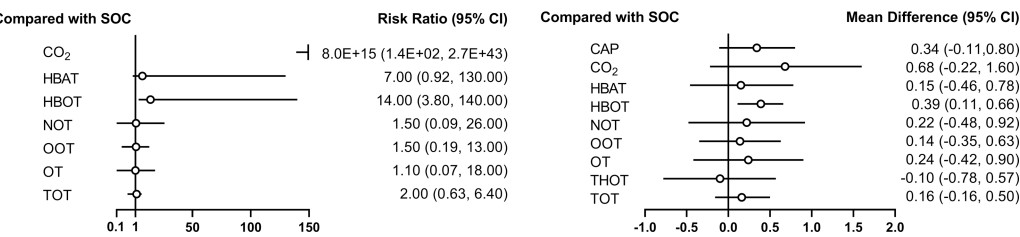

**Figure 5** **Forest plot of gas therapy compared with SOC.** (A) Healing rate; (B) Area reduction rate. HBOT, Hyperbaric oxygen therapy; HBAT, Hyperbaric air therapy; TOT, Topical oxygen therapy; THOT, Topical hyperbaric oxygen therapy; OT, Ozone therapy; OOT, Oxygen-Ozone therapy; $CO_2$, Carbon dioxide; NOT, Nitric oxide therapy; CAP, Cold atmospheric plasma; SOC, Standard of care.

test suggested that $I^2 = 56\%$; therefore, a random-effects model was used. SUCRA analysis (cumulative probability ranking graph Fig. S2B) ranked SOC highest in amputation rates in DFUs (SUCRA = 0.953), followed by HBOT (SUCRA = 0.621). The league table (Table S2B) showed that SOC had a higher amputation rate than HBOT (MD = −0.82, 95% CI [−1.12 to −0.55]), while HBOT had a higher amputation rate than NOT (MD = −19.06, 95% CI [0.63–55.48]). However, no significant differences were observed in amputation rates between HBOT and other gas therapies ($P > 0.05$). Forest plots illustrating amputation rates for each gas therapy compared to SOC are shown in Fig. S3B.

### Adverse events

A total of 14 RCTs examined the incidence of adverse events in DFUs treated with different gas therapies, and the network diagram is presented in Fig. S1C. The heterogeneity test indicated that $I^2 = 31\%$, warranting a fixed-effects model. SUCRA analysis (cumulative probability ranking graph Fig. S2C) ranked HBOT treatment highest for adverse event incidence (SUCRA = 0.832). However, The league table (Table S2C) showed no statistically significant differences in adverse events between HBOT and other gas therapies ($P > 0.05$). Forest plots comparing adverse events for each gas therapy with the SOC are shown in Fig. S3C.

### Quality of life and cost of treatment

Although five RCTs reported the effect of different gas treatments on the quality of life of patients with DFUs, conducting an NMA was not feasible due to the use of different questionnaires across studies. Specifically, two RCTs on HBOT therapy reported conflicting results. One study (*Chen et al., 2017*) found significant improvements in the HBOT group compared with the SOC group based on the Physical Component Summary and Mental Component Summary in the 36-item Short Form Survey Instrument (SF-36) at all time points. The other study (*Abidia et al., 2003*) found no significant differences in SF-36 or the Hospital Anxiety and Depression Scale between the HBOT group and the SOC group. Conversely, two RCTs on TOT yielded consistent findings, with one study (*Anirudh et al., 2021*) reporting improved quality of life in the TOT group based on the wound-QOL questionnaire. The other study (*Frykberg et al., 2020*) showed that patients in the TOT group

achieved the greatest improvement in the well-being component of the Cardiff Wound Impact Schedule-QOL assessment. In addition, an RCT on CAP therapy (*Stratmann et al., 2020*) found that the EuroQOL Five Dimensions Questionnaire and 12-Item Short-Form Health Survey showed comparable changes in average scores in the CAP and SOC groups. Regarding cost-effectiveness, only one RCT evaluated the cost of treatment of HBOT *versus* the SOC (*Abidia et al., 2003*), concluding that HBOT led to significant potential cost-saving, with an average reduction of £2,960 per treated patient compared with the SOC. However, an NMA was not feasible due to the inclusion of only a single RCT.

### Heterogeneity assessment

Heterogeneity was examined for each primary outcome. For ulcer healing rate, the heterogeneity in reticulation comparisons was 60.4% for HBOT *vs.* HBAT and 84.5% for SOC *vs.* HBOT (Fig. S4A). For ulcer area reduction rate, the heterogeneity of reticulation comparisons was 55.6% for SOC *vs.* CAP, 98.2% for SOC *vs.* HBOT, 98.8% for SOC *vs.* OOT, and 84.6% for TOT *vs.* SOC (Fig. S4B). The remaining studies exhibited <50% heterogeneity in reticulation comparisons for the primary outcome metrics.

### Subgroup analysis

In the network analysis of ulcer area reduction rate, an $I^2$ of 86% indicated high heterogeneity, prompting a subgroup analysis. All RCTs were divided into two subgroups according to treatment duration: >6 weeks and ≤6 weeks. The heterogeneity test suggested that $I^2 = 32\%$ in the subgroup with a treatment duration of >6 weeks; thus, the fixed-effects model was used. SUCRA analysis ranked HBOT as the most effective treatment for ulcer area reduction (SUCRA = 0.893). However, in the ≤6-week subgroup, the heterogeneity test indicated an $I^2$ of 91%, suggesting high heterogeneity. Subsequently, a random-effects model was adopted. SUCRA analysis identified $CO_2$ therapy as the most effective for healing rate (SUCRA = 0.829), followed by HBOT (SUCRA = 0.716).

### Publication Bias

To assess the potential publication bias, comparison-adjusted funnel plots were constructed to assess the primary outcomes of the healing and ulcer area reduction rates. The scatter points in the funnel plots were symmetrical and primarily concentrated in the upper section of the funnel, indicating a low risk of significant publication bias. However, some scatter points were outside the 95% CI range (represented by dashed lines) (Fig. 6).

## DISCUSSION

DFUs are one of the common complications in patients with diabetes, and their treatment is complex and challenging. In recent years, gas therapy has received increasing attention as a novel approach for DFU management. To the best of our knowledge, this is the first reticulated meta-analysis to compare the efficacy of different gas-based treatments, including HBOT, TOT, ozone therapy, OOT, $CO_2$ therapy, NO therapy, and CAP therapy for DFUs. In this NMA, we found that HBOT was associated with improved ulcer healing rates and greater ulcer area reduction; however, at the same time, it was associated with a higher amputation rates.
___________________________________________________

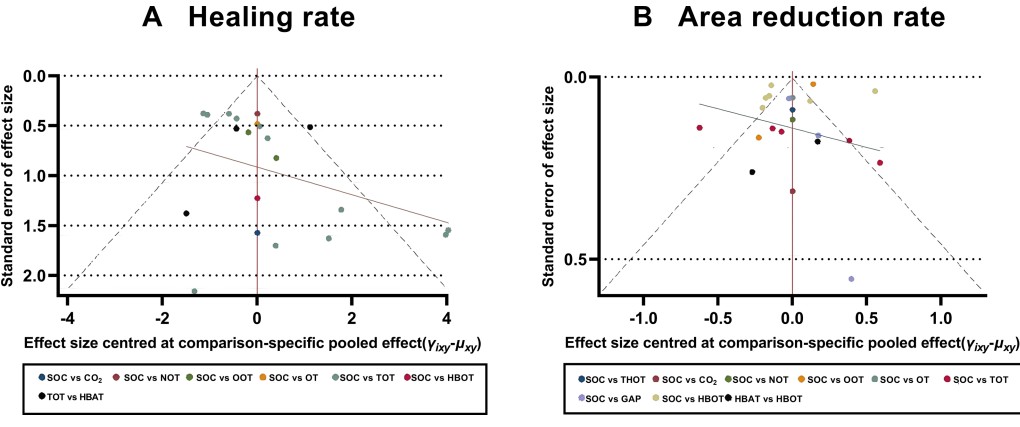

**Figure 6  Comparison-adjusted funnel plots.** (A) Healing rate; (B) Area reduction rate. HBOT, Hyperbaric oxygen therapy; HBAT, Hyperbaric air therapy; TOT, Topical oxygen therapy; THOT, Topical hyperbaric oxygen therapy; OT, Ozone therapy; OOT, Oxygen-Ozone therapy; $CO_2$, Carbon dioxide; NOT, Nitric oxide therapy; CAP, Cold atmospheric plasma; SOC, Standard of care.

Based on a network meta-analysis comparing the efficacy of different gas-based therapies for DFUs, HBOT was the second most effective gas therapy after $CO_2$. Although $CO_2$ therapy ranked higher than HBOT in terms of ulcer healing rate and reduction in ulcer area, this finding was based on a single RCT (*Macura et al., 2020*), which may have resulted in an insufficient sample size or publication bias. The inclusion of only one RCT is a limitation, as a small sample size may not be representative of broader clinical outcomes. In addition, the positive results of $CO_2$ therapy in this single study may have contributed to publication bias. In contrast, although HBOT ranked second in healing rate and ulcer area reduction rate, its efficacy was supported by multiple RCTs, demonstrating more stable and consistent results. HBOT is an emerging therapeutic modality widely used to treat DFUs. The 2023 Edition of the IWGDF Guidelines also recommends considering hyperbaric oxygen as an adjunctive treatment for neuroischemic or ischemic diabetes-related foot ulcers when standard therapies are ineffective and appropriate resources are available (*Chen et al., 2024*). Therefore, despite the high SUCRA ranking of $CO_2$ therapy, the current evidence is insufficient to support definitive clinical recommendations. Thus, we recommend HBOT as the primary gas-based treatment for DFUs. Further large-scale, long-term, and rigorously designed studies are needed to validate the efficacy of $CO_2$ therapy and further compare its efficacy with HBOT.

HBOT promotes wound healing and infection prevention by allowing patients with DFUs to be treated in a high-oxygen-pressure environment. This treatment is usually administered in specialized hyperbaric oxygen chambers, where patients breathe pure oxygen at high concentrations, thereby increasing their systemic oxygen levels (*Sen & Sen, 2021*). Although the precise mechanism of action of HBOT in the treatment of DFUs has still not been fully elucidated, its therapeutic action is believed to involve two pathways. First, HBOT enhances tissue oxygenation by improving the oxygen-delivery ability to injured tissues, thereby meeting the oxygen demands of poorly perfused areas and promoting

wound healing (*Klakeel & Kowalske, 2022*). Second, it promotes neovascularization by stimulating endothelial-type NO synthase, leading to NO production. Subsequently, NO promotes the activation and recruitment of endothelial progenitor cells, stimulating vascular neogenesis and accelerating ulcer healing (*Benincasa et al., 2019*). HBOT may also promote the production of stromal-derived factor-1 and vascular endothelial growth factor by fibroblasts *via* activation of the hypoxia-inducible factor-1α signaling pathway, thereby regulating the angiogenic activity of endothelial cells and promoting neovascularization (*Huang et al., 2020*). Third, HBOT modulates the inflammatory response by inhibiting the growth of numerous pathogenic bacteria, especially anaerobic bacteria, which are unable to survive under hyperbaric oxygen conditions. HBOT also increases free oxygen radical levels and reactive oxygen species (ROS) levels, which help mitigate the inflammation and accelerate wound healing (*De Wolde et al., 2021*). Furthermore, HBOT inhibits the production of pro-inflammatory factors, including interferon-γ, prostaglandins, tumor necrosis factor-α, and interleukin-6, while promoting neutrophil apoptosis at the site of infection, thereby facilitating the resolution of inflammation (*Memar et al., 2019*).

Although our NMA found that HBOT is more effective than other gas-based treatment modalities for DFUs, the associated amputation rate and incidence of adverse events were notably high and should not be overlooked. In the present study, the amputation rate associated with HBOT was second only to that observed with SOC. Several factors may contribute to this elevated rate. First, many of the included RCTs on HBOT lacked blinding, introducing potential bias. Patients with more severe or advanced ulcers were more likely to receive HBOT, which may have skewed the outcomes, highlighting the importance of careful patient selection. Second, the timing of treatment plays a pivotal role; early intervention with HBOT may improve outcomes, whereas delayed administration may be less effective in preventing complications such as amputation. Third, the frequency and duration of treatment influences efficacy, as shorter or insufficient treatment may limit the benefits of HBOT. Lastly, the chronicity of the ulcer is a key determinant, given that long-standing ulcers may be less responsive to any treatment modality, including HBOT. These factors necessitate careful patient selection and individualized treatment protocols to maximize the benefits of HBOT while minimizing associated risks. Taken together, it is evident that HBOT may be less effective in certain disease states or patient populations, suggesting that its indications and optimal usage conditions must be re-evaluated. Our study found that HBOT had the highest-ranked rate of adverse events, even higher than that in the SOC group. In contrast, other gas treatments ranked lower in both amputation and adverse event rates than HBOT and were largely considered unrelated to the treatment measure. Although no statistically significant difference in adverse events was observed between HBOT and other gas therapies, the potential adverse effects of HBOT should not be overlooked. HBOT-related adverse effects were mainly associated with pressure- and oxygen-related complications. The most common adverse events associated with HBOT, as identified in the included RCTs, was an inability to equalize middle ear pressure (*Löndahl et al., 2010*; *Fedorko et al., 2016*; *Santema et al., 2018*). In addition, some patients experienced dizziness (*Löndahl et al., 2010*), pneumatic traumatic tympanic membrane perforation (*Santema et al., 2018*), pneumatic traumatic otitis media

(*Faglia et al., 1996*; *Löndahl et al., 2010*), and oxygen-induced seizures (*Santema et al., 2018*). The high oxygen-pressure environment can create an imbalance between middle ear pressure and external atmospheric pressure, potentially causing pain or discomfort, especially in patients unable to equalize pressure *via* the eustachian tube. However, such pressure-related complications are typically mild and can be prevented or minimized by teaching auto-inflation techniques or inserting tympanostomy tubes (*Camporesi, 2014*). Furthermore, oxygen can induce neurological, respiratory, and visual system damage, thereby warranting attention. Therefore, when considering HBOT for treating DFUs, it is necessary to carefully assess patients' underlying conditions, carefully determine treatment indications, and implement strict procedural controls to minimize the risk of adverse events.

In addition to HBOT, other gas treatments, especially TOT, show potential in treating DFUs. The number of RCTs on TOT included in this NMA was second only to HBOT, indicating that TOT is currently gaining increased attention in treating DFUs. Moreover, the IWGDF 2023 Guidelines recommend local oxygen as an adjunct therapy to treat DFUs (*Chen et al., 2024*). TOT can ameliorate oxygen deficiency by directly delivering oxygen to the wound bed without relying on the vascular or respiratory system (*De Smet et al., 2017*). It may promote wound healing *via* pathways associated with antimicrobial effects, collagen production, and epithelial cell migration (*Kaufman et al., 2018*). Although current guidelines do not recommend ozone, $CO_2$, NO, and CAP therapies as adjuvant treatments for DFUs (*Chen et al., 2024*), these modalities have distinct therapeutic potential. While evidence is limited, future studies should explore their efficacy through rigorously designed multicenter RCTs with standardized protocols. Subgroup analyses stratified by ulcer severity, glycemic control status, and peripheral arterial disease comorbidity may help identify responsive patient populations. Furthermore, dose-optimization studies are needed to establish evidence-based treatment parameters for gas concentration, application frequency, and duration, particularly for ozone and NO therapies, where pharmacokinetic profiles remain undefined.

We acknowledge the methodological limitations of the included RCTs, which may have affected the reliability of our findings. Most studies lacked a double-blind design and had a high risk of bias. Although we addressed outcomes with higher heterogeneity using a random-effects model, this approach may still introduce bias. Notably, the high risk of bias and absence of blinding could systematically overestimate treatment effects, particularly for subjective outcomes such as ulcer healing rates. This may have influenced the relative rankings of interventions, potentially inflating the apparent efficacy of HBOT compared to SOC. A high degree of heterogeneity was found in the network comparisons of certain interventions in the primary outcome indicators. For example, in terms of the ulcer healing rate, the heterogeneity of the net comparison between SOC and HBOT was 84.5%. A similar pattern was observed for the ulcer area reduction rate, with 98.2% heterogeneity in the reticulation comparisons between SOC and HBOT, 98.8% heterogeneity in the reticulation comparisons between SOC and OOT, and 84.6% heterogeneity in the reticulation comparisons between TOT and SOC. These substantial heterogeneity levels may reflect variations across studies, including

differences in patient demographics, treatment adherence, external influencing factors, treatment regimens, duration, or assessment methods. Therefore, additional rigorous, high-quality, large-sample RCTs are needed to validate these findings. To determine the sources of heterogeneity in study results, a subgroup analysis was performed on the highly heterogeneous primary outcome indicator, the ulcer area reduction rate. The RCTs were categorized into two subgroups based on treatment duration: >6 weeks and ≤6 weeks. In the subgroup with the treatment duration of >6 weeks, the test of heterogeneity in the reticulation analysis indicated a substantial reduction in the ulcer area heterogeneity ($I^2 =$ 32%) compared to the overall analysis. SUCRA analysis revealed that HBOT had the best efficacy in promoting reduction of the ulcer area. In contrast, the subgroup analysis for treatment duration of up to 6 weeks showed persistently high heterogeneity ($I^2 = 91\%$). These findings suggest that HBOT administered for >6 weeks may be more effective in reducing ulcer area in DFUs. Assessment of publication bias using comparison-adjusted funnel plots revealed a relatively low risk, as the scatter points were largely symmetrical. However, some points fell outside the 95% CI range, suggesting a potential tendency for positive study results to be preferentially accepted and published. If present, this bias could artificially enhance the perceived efficacy of newer or more extensively researched interventions (*e.g.*, HBOT) compared to standard therapies, as negative or neutral trials may remain unpublished. Such bias could compromise the validity of treatment rankings and potentially mislead clinical decision-making.

Rigorous efforts were made to extract data on the quality of life of patients and the cost of gas therapy; however, only five RCTs reported quality of life outcomes. Furthermore, due to variations in the assessment tools used across the studies, an NMA comparing the effectiveness of various gas therapies in improving quality of life of patients could not be performed. Nevertheless, these studies provided preliminary insights into the potential benefits of gas therapy. In particular, some studies reported positive effects of HBOT (*Chen et al., 2017*) and TOT (*Frykberg et al., 2020*; *Anirudh et al., 2021*) on quality of life, whereas the effects of CAP therapy were less pronounced. Despite methodological differences, these findings highlight the need for further research to systematically evaluate and compare the impact of different gas therapies on quality of life of patients. Regarding treatment costs, very few studies have addressed this aspect, with only one RCT providing data on HBOT-related costs. This study (*Abidia et al., 2003*) suggested potential cost-saving benefits of HBOT but did not account for the substantial initial investment required to establish a hyperbaric oxygen facility. However, in settings where hyperbaric facilities are already available for other medical applications, HBOT may be a cost-effective option if it can significantly improve wound healing in patients with DFUs (*Chen et al., 2024*).

Our NMA has several limitations. First, the results may have been influenced by variations in treatment duration, treatment frequency, degree of ischemia, and the definition of ulcer healing. Second, due to data limitations in the included RCTs, the long-term efficacy and safety of gas therapy for the treatment of DFUs could not be analyzed. Third, more than half of the included studies lacked a double-blind design or exhibited a high risk of bias, necessitating cautious interpretation of the findings. Lastly, some gas therapies were evaluated in only a small number of RCTs, with some

interventions supported by a single trial, potentially leading to insufficient sample sizes and publication bias. Thus, further high-quality studies are needed to validate the efficacy of these interventions.

## CONCLUSIONS

Although HBOT demonstrated certain advantages over other gas therapies in promoting DFU healing, its association with a higher amputation rate warrants careful consideration. Therefore, clinicians should comprehensively assess patient-specific conditions, weigh the benefits and risks, and exercise caution when implementing HBOT, particularly in resource-limited settings. Given the limited number and quality of included studies, these findings require validation through high-quality research. In addition, future studies should further explore factors influencing HBOT efficacy and optimize treatment protocols to enhance safety and therapeutic outcomes.

## ACKNOWLEDGEMENTS

We would like to thank MogoEdit for its English editing during the preparation of this manuscript.

### Funding

This project was supported by the Scientifc Research Fund of Chengdu Fifth People's Hospital (No.GSPZX2022-19). The funders had no role in study design, data collection and analysis, decision to publish, or preparation of the manuscript.

### Grant Disclosures

The following grant information was disclosed by the authors:
The Scientifc Research Fund of Chengdu Fifth People's Hospital:  No. GSPZX2022-19.

### Competing Interests

The authors declare there are no competing interests.

### Author Contributions

- Jing Yang conceived and designed the experiments, performed the experiments, analyzed the data, prepared figures and/or tables, authored or reviewed drafts of the article, and approved the final draft.
- Peng Ning conceived and designed the experiments, analyzed the data, prepared figures and/or tables, and approved the final draft.
- Jiali Huang performed the experiments, prepared figures and/or tables, and approved the final draft.
- Hong Ouyang performed the experiments, prepared figures and/or tables, and approved the final draft.

- Jiaxing Zhang performed the experiments, prepared figures and/or tables, and approved the final draft.
- Fan Yang performed the experiments, prepared figures and/or tables, and approved the final draft.
- Hongyi Cao performed the experiments, prepared figures and/or tables, supervised the study, and approved the final draft.
- Fan Zhang conceived and designed the experiments, authored or reviewed drafts of the article, and approved the final draft.

## Data Availability

The raw measurements are available in the Supplementary File.

## Supplemental Information

Supplemental information for this article can be found online at http://dx.doi.org/10.7717/peerj.19571#supplemental-information.

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
