# Peer review of "Comparative efficacy of gas therapy for diabetic foot ulcers using network meta-analysis"

_PeerJ, doi:10.7717/peerj.19571_

## Round 0.1 · original submission · Major Revisions

Dear author, following the instructions from reviewers 1 and 5, we suggest major revisions respond to both. We look forward to improving our work and producing a high-quality paper after the suggested changes. Thank you very much.

Reviewer 1 ·

Basic reporting

English language is good,
In the introduction section, some more elaboration can be added on all the types of therapies if there is no restriction on the word count.
Well structured paper.

Experimental design

There is a suggestion given in the pdf file.

Validity of the findings

It is a good study that combines all the therapies for DFU healing. All the possible states have been used.

Annotated reviews are not available for download in order to protect the identity of reviewers who chose to remain anonymous.

Reviewer 2 ·

Basic reporting

Language, explained methodolgy and structure of manuscript is clear and proffessional.
I do not have a deep experience in statistical methodology of such meta analysis. A proffessional on matter should examine the validity of used approaches.

Experimental design

Regarding design of meta-analyses. I am not expert in that regard, However, including all types HBOTs as homogenious treatment caught my attention since for such long timespan, there should be very spesific distinctions among different applications.

Validity of the findings

Main problem of this study is its data interpretation and conclusion, which are way bolder than what we know currently and what this meta-analyses could find. Such statements are not backed up with such findings. There is a mismatch in cause and effect relationship. Did HBOT cause more mortality than other treatments or did studies applied HBOT in more severe ulcers that are more likely to have amputation? I think the second one is more likely. And on the other hand, HBOT is widely known as quiet safe treatment, which is again contradicting with auhors' conclusion. I think such mis/overstatements stemmed from wrong interpretation of acquisited data.

Reviewer 3 ·

Basic reporting

Clear

Experimental design

Method is Clear

Validity of the findings

There is Novelty in this study.

Reviewer 4 ·

Basic reporting

- Clear and unambiguous, professional English used throughout.
The text is written in clear, precise, and technically accurate English, incorporating appropriate scientific terminology.

- Literature references, sufficient field background/context provided.
The introduction is well-structured and comprehensive, addressing the key topics relevant to the review while incorporating recent publications in the field.

- Professional article structure, figures, tables. Raw data shared.
The article follows the "standard sections" outlined in our Instructions for Authors, with relevant figures that align with the review's content, presented in appropriate resolution, and accurately described and labeled.

Experimental design

This review on the use of gas therapies for diabetic foot ulcers (DFUs) is highly relevant to the journal, as it addresses a topic of cross-disciplinary interest in various fields, such as diabetes, endocrinology, general medicine, and surgery. Diabetic foot ulcers represent one of the most common and severe complications of diabetes, and treating them remains a challenge in clinical practice due to the complexity of wound healing in patients with this condition. The focus on gas-based therapies, such as hyperbaric oxygen therapy (HBOT), ozone therapy, CO2 therapy, among others, holds great potential for improving outcomes in the treatment of diabetic foot ulcers. This review provides updated evidence on the comparative efficacy of these treatments, which is essential to guide clinical decisions and enhance therapeutic protocols for patients with DFUs. Furthermore, the relevance of this topic is amplified by the high prevalence of diabetes worldwide, making diabetic foot ulcers a significant public health issue. The review, based on a meta-analysis of recent studies, provides valuable data that can influence clinical practice and future research, favoring a more efficient and safer approach to the treatment of diabetic ulcers. Therefore, this article is pertinent to the journal because it contributes to advancing knowledge in the treatment of a critical diabetes complication, improving both patients' quality of life and health outcomes, and aligns with the journal’s focus on clinically and scientifically relevant topics.

Validity of the findings

Impact: given the clinical importance of the topic and the fact that the article reviews gas therapies for the treatment of diabetic foot ulcers, it can be considered a novel contribution to the existing literature. The combination of various gas therapy modalities in a comparative analysis has the potential to significantly influence clinical practices and future research.

Significant Replication: the article promotes meaningful replication by clearly presenting the justification and benefits of gas therapies for diabetic foot ulcers. The inclusion of a meta-analysis of randomized controlled trials (RCTs) enables the replication of its findings through additional studies that use similar methodologies. Furthermore, the review outlines the limitations of the included studies, which facilitates critical interpretation and controlled replication of these therapeutic approaches.

Conclusions: the conclusions are well-expressed and clearly linked to the original research question regarding the comparative efficacy of gas therapies for diabetic foot ulcers. The article appropriately demonstrates how the results of the meta-analysis support the advantage of certain therapies, such as hyperbaric oxygen therapy (HBOT), compared to other options. However, the conclusions are limited to the results of the studies included in the review, which is appropriate given that the study is based on the available evidence. There is no excessive extrapolation beyond what the data can support, ensuring that the conclusions are responsible and evidence-based.

·

Basic reporting

The study is investigating an interesting subject, as the authors evaluate the effectiveness of different gas therapy modalities, particularly hyperbaric oxygen therapy (HBOT) and topical oxygen therapy (TOT), in the treatment of diabetic foot ulcers (DFUs) through a network meta-analysis (NMA). The study aims to compare these gas therapies based on healing outcomes, including healing rate, wound area reduction, amputation rate, and adverse events, in order to provide evidence-based insights for optimizing DFU treatment strategies. I have thoroughly reviewed the article some aspects of the paper should be addressed for improvement:
1. The current title is little bit long, distracted. It would benefit from revision by the authors to make it in appropriate length, more concise, and informative, capturing the reader’s attention with clear and appealing language.
2. Kindly add a highlight bullets in the research will be very helpful to attract the readers attention and provide the brief data about the study which reflects the importance of the research presented.
3. Kindly in the “Abstract” section needs to be revised by the authors and re-editing in a better presentable manner without skipping the ideas to comply with the presented idea of the manuscript and more organized to present the idea of the research in a better, several week points which needs to be enhanced in this section and reflects on other sections in the article.
4. The “Introduction” section effectively highlights the clinical significance of DFUs, emphasizing their impact on healthcare systems and the need for improved treatment strategies. However, it still needs to be revised and enhanced in each of the following:
While the Authors acknowledges traditional treatments, it does not elaborate on their limitations (e.g., antibiotics, debridement, wound dressings). A brief comparison with gas therapies could strengthen the study's justification.
Kindly expand the current treatment challenges and limitations to better justify the focus on gas therapies.
The Authors mentions the use of network meta-analysis (NMA) but does not explain why this method is superior to conventional meta-analysis for evaluating DFU treatments. A brief mention of its advantages would improve clarity
The Authors states the objective but does not provide a clear hypothesis or expected findings. Stating whether HBOT or TOT is hypothesized to be more effective would provide a stronger research direction.
5. kindly read some of these useful previous researches with various techniques toward your study (Investigating HBOT for the DFUs) which could be helpful in your “Introduction and Literature sections”, feel free to use them in your study or not after reading them:
Fahmy, Siham M., et al. "Hyperbaric oxygen therapy for healing diabetic lower extremity ulcers." 2020 12th International Conference on Electrical Engineering (ICEENG). IEEE, 2020.
Wadee, Amir N., et al. "The influence of low-intensity laser irradiation versus hyperbaric oxygen therapy on transcutaneous oxygen tension in chronic diabetic foot ulcers: a controlled randomized trial." Journal of Diabetes & Metabolic Disorders 20 (2021): 1489-1497.
Zhang, Zhiming, et al. "Efficacy of hyperbaric oxygen therapy for diabetic foot ulcers: an updated systematic review and meta-analysis." Asian journal of surgery 45.1 (2022): 68-78.
Tao, Lihua, and Xiao Yuan. "Efficacy and safety of hyperbaric oxygen therapy in the management of diabetic foot ulcers: A systematic review and meta‐analysis." International Wound Journal 21.3 (2024): e14507.
Sharma, Rakesh, et al. "Efficacy of hyperbaric oxygen therapy for diabetic foot ulcer, a systematic review and meta-analysis of controlled clinical trials." scientific reports 11.1 (2021): 2189.

Experimental design

assessment strategy, and software details would enhance clarity, transparency, and reproducibility, needs to be revised and re-edit regarding the following recommendations:
The section does not clearly justify why Bayesian models were chosen over frequentist models in the NMA, kindly add a brief explanation of its advantages would add clarity.
The Authors mentions Bayesian modeling but does not specify which software (e.g., R, STATA, or WinBUGS) was used for data analysis. This information is crucial for reproducibility.
No rationale is given for the specific subgroup analysis criteria (e.g., patient demographics, wound severity).
The methodology does not explicitly state how publication bias and study heterogeneity were addressed, aside from sensitivity analysis.
It was unclear whether the Risk of Bias (RoB 2.0) tool or GRADE framework was used to assess study quality.

Validity of the findings

7. The “Result” is well-structured, with clear data presentation and a strong comparative analysis using NMA. However, it can be improved in the following areas:
The section presents statistical outcomes without sufficient interpretation, leaving some results without clinical context.
While credible intervals and rankings are included, the study does not consistently highlight which results are statistically and clinically significant.
Some results appear numerically different but are not statistically significant, which should be explicitly stated.
The reported values for treatment rankings sometimes lack confidence/credible intervals, making it difficult to assess uncertainty around estimates. Kindly, ensure that all comparisons include appropriate uncertainty measures to reinforce credibility.
The Authors primarily focuses on positive findings but does not discuss any negative or inconclusive results. Addressing potential limitations or contradictory findings would provide a more balanced and transparent analysis.

8. The “Discussion” section is well-structured, with strong comparisons to previous studies and a clear emphasis on clinical relevance. However, it can be strengthened by:
While the discussion summarizes the results well, it does not always provide detailed explanations for unexpected or conflicting findings. There is limited discussion on the biological or mechanistic reasons behind treatment differences.
Although limitations are acknowledged, the discussion could elaborate more on confounding variables, such as patient demographics, treatment adherence, or external factors influencing outcomes.
The study briefly mentions publication bias and heterogeneity, but it does not thoroughly explore how these biases may have influenced the final treatment rankings. More discussion on study selection bias and its effect on NMA rankings would add credibility to the conclusions.
While the study suggests areas for further research, it lacks specific recommendations on how future studies should be designed to overcome existing gaps. More precise suggestions, such as targeted clinical trials or subgroup analyses, would strengthen this section.

Additional comments

9. Kindly revise the “Reference” section and update it, to cover your study`s background and literature review. Additionally, to be in the latest and nearest year to your research, as some reference needs to be replaced with the updated researches, such as: (Ref #23-1992 / Ref #24-1996 / Ref #25-2003 / Ref #26-2003 / Ref #27-2008 / Ref #28-2010 /…).
10. The English language in this manuscript requires considerable improvement, as it contains several typographical errors and overly long sentences with mixed ideas,which reduce clarity and impact. Some sentences are long and contain excessive technical jargon. Simplifying certain passages or breaking them into smaller sections could improve readability.
I recommend the authors revise the manuscript with more concise and well-structured sentences and seek professional English proofreading before resubmitting it.
In conclusion, I commend the authors for their hard work and efforts in preparing this manuscript. However, I believe further refinement is needed to improve the organization of ideas and to more clearly present the study's findings with additional detail and clearer illustrations.
Best regards and good luck,

---

## Round 0.2 · accepted · Accept

All the comments were addressed.

·

Basic reporting

No additional comments

Experimental design

No additional comments

Validity of the findings

No additional comments